# Association of ankle-brachial index with cognitive decline in patients with lacunar infarction

Masahiro Nakamori[1,2]*, Hayato Matsushima[2], Keisuke Tachiyama[1,2], Yuki Hayashi[1,2], Eiji Imamura[2], Tatsuya Mizoue[3], Shinichi Wakabayashi[3]

**1** Department of Clinical Neuroscience and Therapeutics, Hiroshima University Graduate School of Biomedical and Health Sciences, Hiroshima, Japan, **2** Department of Neurology, Suiseikai Kajikawa Hospital, Hiroshima, Japan, **3** Department of Neurosurgery, Suiseikai Kajikawa Hospital, Hiroshima, Japan

* mnakamori1@gmail.com

**Data Availability Statement:** All relevant data are within the paper and its Supporting Information files.

## Abstract

Atherosclerosis is an important risk factor for cognitive decline. This study aimed to investigate the relationship of ankle-brachial pressure index (ABI) and brachial-ankle pulse wave velocity (baPWV) with cognitive function in patients with lacunar infarction. We included records of consecutive patients with their first-ever acute stroke and a diagnosis of lacunar infarction through magnetic resonance imaging (MRI) from July 1, 2011 to December 31, 2018. We excluded patients diagnosed with dementia, including strategic single-infarct dementia, before or after stroke onset. Moreover, we excluded patients with one or more microbleeds, severe white matter lesions, or severe medial temporal atrophy on MRI. For ABI, we used the lower ankle side and divided the results into ABI < 1.0 and ABI $\geq$ 1.0. For baPWV, we used the higher ankle side and divided the results into two groups based on the median value of the participants. We analyzed 176 patients with stroke (age 72.5 ± 11.4 years, 67 females). The median score on the Mini-Mental State Examination (MMSE) was 27. The number of patients with ABI < 1.0 was 19 (10.8%). Univariate analysis revealed that the MMSE score was associated with age, body mass index, education, chronic kidney disease, periventricular hyperintensity, and ABI < 1.0 (p < 0.10), but not baPWV. Multivariate analysis revealed that body mass index (p = 0.039) and ABI < 1.0 (p = 0.015) were independently associated with the MMSE score. For patients with lacunar infarction, a lower ABI, but not a higher PWV, was associated with cognitive decline.

## Introduction

In an aging society, cognitive decline, which causes morbidity and mortality, is among the most critical issues with respect to population health, care, and medical economics [1, 2]. Lifestyle habits, including vascular risk factors, contribute to stroke and vascular cognitive impairment [3, 4]. Additionally, vascular disease is an important modifiable risk factor for clinically diagnosed Alzheimer's dementia and related dementias [5, 6]. Moreover, atherosclerosis and cerebral circulation are important factors for cognitive decline.

**Funding:** The authors received no specific funding for this work.

**Competing interests:** The authors have declared that no competing interests exist.

The ankle-brachial pressure index (ABI) and pulse wave velocity (PWV), which reflect arterial stenosis and arterial stiffness, respectively, are commonly used for objective non-invasive assessment of atherosclerosis. ABI is the ratio of the ankle and brachial systolic blood pressure. It is associated with arterial stenosis severity or leg occlusion, which results in a lower ABI. Peripheral artery disease is diagnosed using an ABI < 0.9. Generally, the ankle blood pressure is higher than the brachial blood pressure; moreover, ABI < 1.0 is indicative of arterial sclerosis or stenosis to some extent. Lower-leg atherosclerosis also represents a similar pathology in other arterial systems, including cerebral circulation [7, 8]. On the other hand, PWV is measured between two sites along the arterial system. This reflects arterial stiffness, which results in a higher baPWV. There are two main measurements; namely, carotid-femoral PWV (cfPWV) and brachial-ankle PWV (baPWV).

There have been previous population-based studies on the association between ABI/PWV and cognitive function [3, 9, 10]. In community-dwelling older populations, a lower ABI, but not a higher baPWV, is an independent risk factor for cognitive impairment [11]. However, other cohort studies have reported higher cfPWV values in individuals with vascular dementia than in those without dementia [12]. Additionally, the cfPWV value is inversely associated with measures for cognitive function, including the Mini-Mental State Examination (MMSE) score [13–15]. There have been varying reports regarding the association between artery assessment and cognitive decline. Diagnosis and treatment of atherosclerosis are crucial for preventing stroke and cognitive decline.

We aimed to investigate the relationship of ABI and baPWV with cognitive function. In this study, we focused on the first-ever acute stroke and a diagnosis of lacunar infarction because the basic mechanism of stroke differs among stroke subtypes. Additionally, in other stroke types, i.e., stroke types other than lacunar infarction, the stroke lesions usually involve the cerebral cortex, which itself can affect the cognitive function. In addition, we had to exclude cases of neurodegenerative diseases. Therefore, we tried to exclude cases of cerebral microbleeds (CMBs) and white matter lesions (WMLs). CMBs, especially lobar type CMBs, are associated with amyloid pathology. Severe WMLs are sometimes associated with neurodegenerative diseases such as leukoencepalopathy. Few reports have assessed the association between atherosclerosis and cognitive decline by excluding such factors. In this study, we investigated the relationship of ABI and baPWV with cognitive function in patients with first-ever lacunar infarction and without CMBs and WMLs.

## Materials and methods

### Ethics

The study protocol was approved by the ethics committee of Suiseikai Kajikawa Hospital (approval number 2019–07) and was performed in accordance with the national government guidelines based on the 1964 Declaration of Helsinki. By the ethics committee of Suiseikai Kajikawa Hospital, the requirement for written informed consent was waived owing to the retrospective nature of this study. Moreover, upon admission, the included patients consented for their data to be used for future studies.

### Participants

We retrospectively included consecutive patients admitted with a first-ever acute stroke diagnosed with lacunar infarction from July 1, 2011 to December 31, 2018. Lacunar infarction was determined according to the criteria of the Trial of Org 10172 in Acute Stroke Treatment [16]. We excluded patients diagnosed with dementia, including strategic single-infarct dementia, before or after stroke onset. Dementia was diagnosed using the 10th revision of the

International Statistical Classification of Diseases and Related Health Problems. Diagnosis was confirmed by two stroke neurologists (HM and EI). Moreover, we excluded patients with one or more CMBs or severe WMLs on magnetic resonance imaging (MRI) because they might be confounding factors for ABI/PWV. MRI was performed using a 1.5T scanner (Avanto, Siemens Medical Systems, Erlangen, Germany) or a 3.0T scanner (Spectra, Siemens Medical Systems, Erlangen, Germany). The imaging protocols were the same for the two MRI groups. We investigated the difference in background characteristics between the two groups and found no significant difference. Gradient-echo T2*-weighted MRI (GRE) was performed to evaluate the presence of CMBs. CMBs were defined as homogeneous round lesions with diameters ≤ 10 mm, which were characterized by signal intensity loss, as shown on GRE. Based on the appearance or clinical history, lesions exhibiting signal intensity loss in the globus pallidus or subarachnoid space and diffuse axonal injury were excluded [17]. The severity of WMLs (deep and subcortical white matter hyperintensity [DSWMH] and periventricular hyperintensity [PVH]) was rated visually on fluid-attenuated inversion recovery images using the Fazekas scale (DSWMH: grade 1, punctuate; grade 2, early confluence; and grade 3, confluent; and PVH: grade 1, caps or lining; grade 2, bands; and grade 3, irregular extension into the deep white matter) [18]. Patients with WMLs (DSWMH or PVH) of grades 3 were assigned to the severe WML groups. In addition, according to a previous report, we evaluated the presence of silent lacunar lesions and graded the number of lacunae as follows: grade 0, absent; grade 1, 1 to 2 lacunae; grade 2, 3 to 5 lacunae; and grade 3, 6 or more lacunae [19]. Two stroke neurologists (MN and KT) graded the patients after consensus.

## Data acquisition

MMSE scores were recorded and ABI/baPWV measurements were performed within 3 days of admission for all patients [20]. The accuracy of the method of ABI/baPWV measurement has been validated previously [21]. ABI/baPWV measurements were performed using BP-203RPE III (OMRON HEALTHCARE Co., Ltd., Kyoto, Japan). For ABI, we used the lower ankle side and the patients were categorized into patients with ABI < 1.0 or ABI ≥ 1.0 because ABI < 1.0 is a risk factor for arterial sclerosis and mortality [7, 8]. For baPWV, we used the higher ankle side and the patients were categorized into two groups based on the median value of the participants. In addition, we performed linear analysis for baPWV.

We recorded baseline clinical characteristics, including age, sex, body mass index (BMI), duration of education, complications (hypertension, diabetes mellitus, dyslipidemia, and chronic kidney disease), current smoking, habitual drinking, and medication before admission (antihypertensive and antidiabetic drugs). The severity of stroke was evaluated using the National Institutes of Health Stroke Scale (NIHSS) [22]. In addition to obtaining the medical history, we identified relevant risk factors from a self-reported medical history or inferred from medications prescribed by the primary physician. The criteria for hypertension, diabetes mellitus, and dyslipidemia were previously defined [23].

## Statistical analysis

We calculated the required sample size for this study according to previous studies that compared MMSE scores with ABI/baPWV or CMBs [11, 24]. Based on an alpha level of 0.05 and power of 0.80, we estimated that we would require at least 128 participants. Univariate analysis was used to investigate the association of MMSE scores with several factors. Subsequently, multivariate analysis was performed to estimate and test the independent effects of selected factors on MMSE score. Each of those factors was determined from univariate analysis if the p value was 0.10 or less. In multivariate analysis, the least squares test was performed with the

selected factors, which were entered simultaneously. For multiple comparisons, the data were analyzed using one-way analysis of variance (ANOVA), followed by post-hoc Tukey's honestly significant difference (HSD) test, with Bonferroni correction. Moreover, the MMSE score was divided into 11 sub-scores (orientation of time, orientation of place, registration, serial sevens, delayed recall, designation, repetition, commands, sentence comprehension, sentence writing, and graphic replication) for statistical analysis. Data were expressed as the mean ± standard deviation or median (25% interquartile range [IQR]–75% IQR) for continuous variables; moreover, frequencies and percentages were presented for discrete variables. Statistical analyses were performed using JMP 15 statistical software (SAS Institute Inc., Cary, NC, USA). Between-group comparisons were performed using ANOVA. Statistical significance was set at $p < 0.05$.

## Results

A total of 826 patients were diagnosed with lacunar infarction; of these, 468 patients were admitted for their first-ever stroke. We excluded 43 patients without MRI data, 25 patients without MMSE scores, and 127 patients diagnosed with dementia, including strategic single-infarct dementia (n = 3), before or after stroke onset. We excluded five patients because they did not undergo ABI/baPWV evaluation. Moreover, we excluded 92 patients with CMBs or severe WMLs. Ultimately, we analyzed 176 patients with stroke (age: 72.5 ± 11.4 years, 67 females; Fig 1). Regarding the silent lacunar lesion, all patients were judged as having grade 3 or less lesions. In this study, no patient had symptomatic peripheral artery disease and required percutaneous transluminal angioplasty. The systolic blood pressure in all patients was maintained under 220 mmHg. No patient required antihypertensive medications such as intravenous calcium channel blockers at the time of parameter measurement within 3 days of admission. None of the patients showed altered consciousness. No new psychiatric drugs, including sleeping pills, were added within 3 days from admission to the recording of MMSE scores. The median MMSE score was 27 (26–29). Table 1 shows the background characteristics of the patients. Data pertaining to systolic and diastolic blood pressure in the four limbs and bilateral ABI and baPWV are shown in S1 Table. We investigated the association between the laterality of ABI and the MMSE score but found no association. We evaluated the influence of stroke severity on cognitive function or ABI/baPWV. Linear regression analyses showed no association between NIHSS and MMSE scores (p = 0.108). Further, there was also no association between NIHSS scores and ABI or baPWV (p = 0.376, p = 0.233, respectively). We considered patients with NIHSS score 0 as almost normal controls and analyzed the effects; there was no significant relationship with the MMSE score, ABI, or baPWV (p = 0.646, p = 0.546, p = 0.132, respectively).

The median baPWV was 2019 cm/s. The patients were categorized into the following four groups: patients with ABI $\geq$ 1.0 and baPWV $\leq$ 2019 cm/s; ABI $\geq$ 1.0 and baPWV > 2019 cm/s; ABI < 1.0 and baPWV $\leq$ 2019 cm/s; or ABI < 1.0 and baPWV > 2019 cm/s. S2 Table presents the characteristics of each group. Fig 2 shows the MMSE scores for each group. ANOVA showed that the MMSE scores were directly and inversely correlated with ABI and baPWV, respectively (p < 0.05). Bonferroni correction and Tukey's HSD tests revealed that the MMSE scores for the group with ABI < 1.0 and baPWV > 2019 cm/s were significantly lower than those for the group with ABI $\geq$ 1.0 and baPWV $\leq$ 2019 cm/s.

We investigated the association of MMSE scores with the factors listed in Table 1. Univariate analysis revealed that the MMSE score was associated with age, BMI, education level, chronic kidney disease, and ABI < 1.0 (p < 0.10), but not with baPWV > 2019 cm/s. The location of lacunar infarction was not associated with the MMSE score. Multivariate analysis with

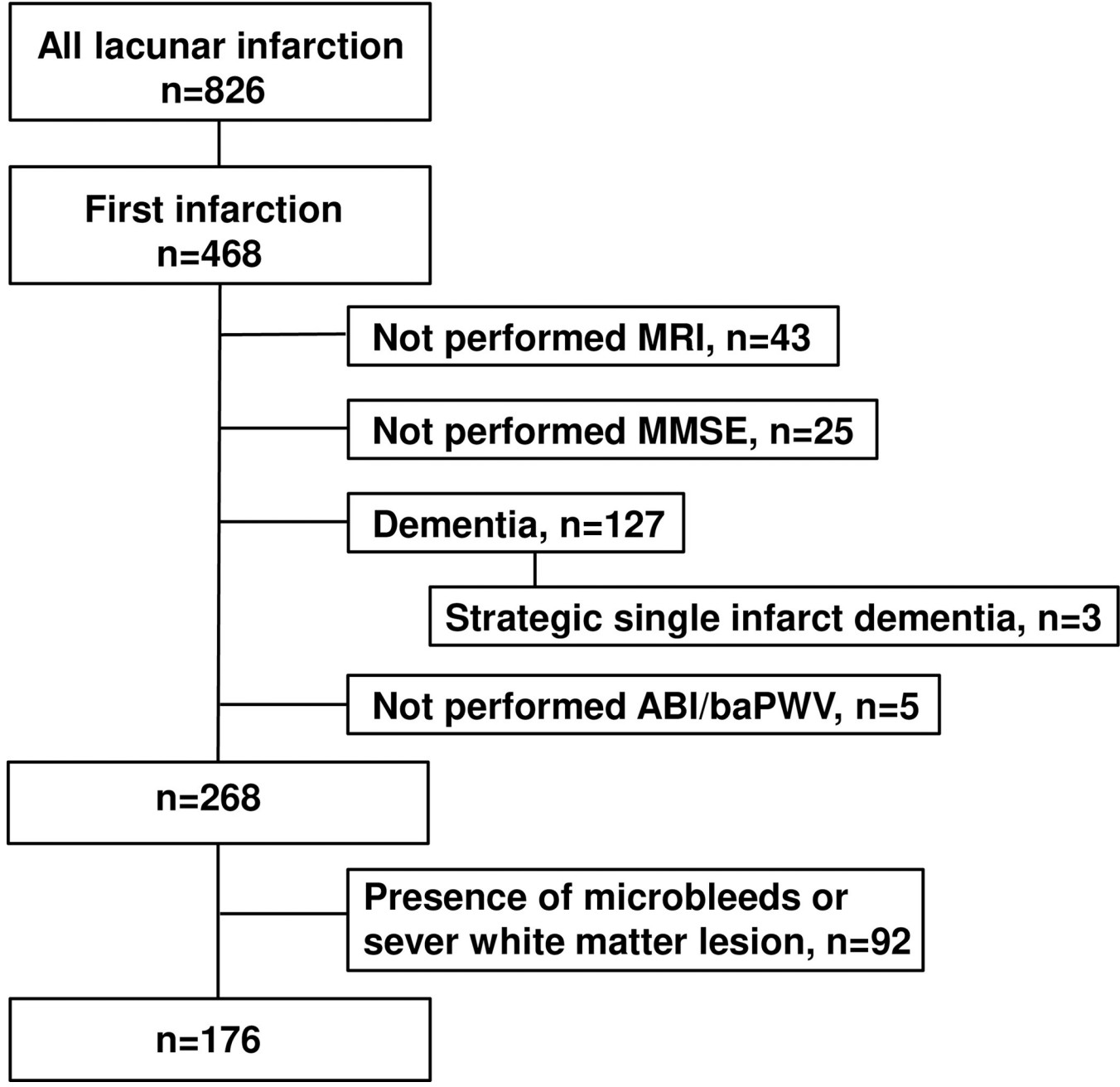

**Fig 1. Flow chart of the inclusion and exclusion criteria.** MRI, magnetic resonance imaging; MMSE, Mini-Mental State Examination.

the identified factors revealed an independent association of BMI (p = 0.032) and ABI < 1.0 (p = 0.014) with the MMSE score (Table 2). We also performed linear analysis for baPWV. Univariate analysis showed that baPWV was associated with MMSE scores (p = 0.011). However, multivariate analysis using age, BMI, education level, and chronic kidney disease as covariates showed that baPWV was not significantly associated with MMSE scores (p = 0.117).

Subsequently, we compared the MMSE sub-scores between the ABI < 1.0 and ABI ≥ 1.0 groups. The sub-scores for orientation and immediate recall were significantly lower in the

**Table 1. Patients' background.**

| | n = 176 |
|---|---|
| Age, year | 72.5±11.4 |
| Sex (female), n (%) | 67 (38.1) |
| Body mass index, kg/m$^2$ | 23.9±3.7 |
| Education, year | 12.5±2.5 |
| MMSE score, median (IQR) | 27 (26–29) |
| Hypertension, n (%) | 124 (70.5) |
| Diabetes mellitus, n (%) | 44 (25.0) |
| Dyslipidemia, n (%) | 101 (57.4) |
| Chronic kidney disease, n (%) | 45 (25.6) |
| Current smoker, n (%) | 64 (36.4) |
| Habitual drinker, n (%) | 72 (40.9) |
| Antihypertensive drug, n (%) | 111 (63.1) |
| Antidiabetic drug, n (%) | 31 (17.6) |
| NIHSS score, median (IQR) | 2 (1, 3) |
| Location of infarction | |
| Side of the lesion (left), n (%) | 89 (50.6) |
| Corona radiata, n(%) | 52 (29.5) |
| Basal ganglia, n(%) | 10 (5.7) |
| Capsulae internae, n(%) | 38 (21.6) |
| Thalamus, n(%) | 50 (28.4) |
| Brain stem, n(%) | 26 (14.8) |
| MRI findings | |
| DSWMH, median (IQR) | 1 (1, 2) |
| PVH, median (IQR) | 2 (1, 2) |
| Ankle Brachial pressure index | 1.10±0.11 |
| Ankle Brachial pressure index <1.0, n (%) | 19 (10.8) |
| Brachial-ankle pulse wave velocity, cm/s | 2139.3±571.1 |

MMSE, Mini-Mental Scale Examination; IQR, interquartile range; NIHSS, National Institutes of Health Stroke Scale; MRI, magnetic resonance imaging; DSWMH, deep and subcortical white matter hyperintensity; PVH, periventricular hyperintensity.

Data are presented as the mean ± standard deviation, median (25% IQR to 75% IQR), or the number of patients (%).

ABI < 1.0 group than in the ABI ≥ 1.0 group after adjustment for age, BMI, education, and chronic kidney disease (p = 0.012 and 0.011, respectively) (Table 3).

Finally, we added the patients with CMBs and severe white matter lesions (who were excluded based on MRI findings) and reanalyzed again (n = 268). S3 Table shows the background characteristics of these patients. The median baPWV was 2086.5 cm/s and divided the patients based on the median baPWV. We investigated the association of MMSE scores with the factors listed in S3 Table. Univariate analysis revealed that the MMSE score was associated with age, BMI, education, chronic kidney disease, cerebral microbleeds, PVH, and ABI < 1.0 (p < 0.10), but not with baPWV > 2086.5 cm/s. Multivariate analysis revealed an independent association of BMI (p = 0.009) and ABI < 1.0 (p = 0.019) with the MMSE score (S4 Table).

## Discussion

In this study, we focused on patients with lacunar infarction and excluded patients with CMBs and severe WMLs on MRI. We observed that a lower ABI was significantly associated with

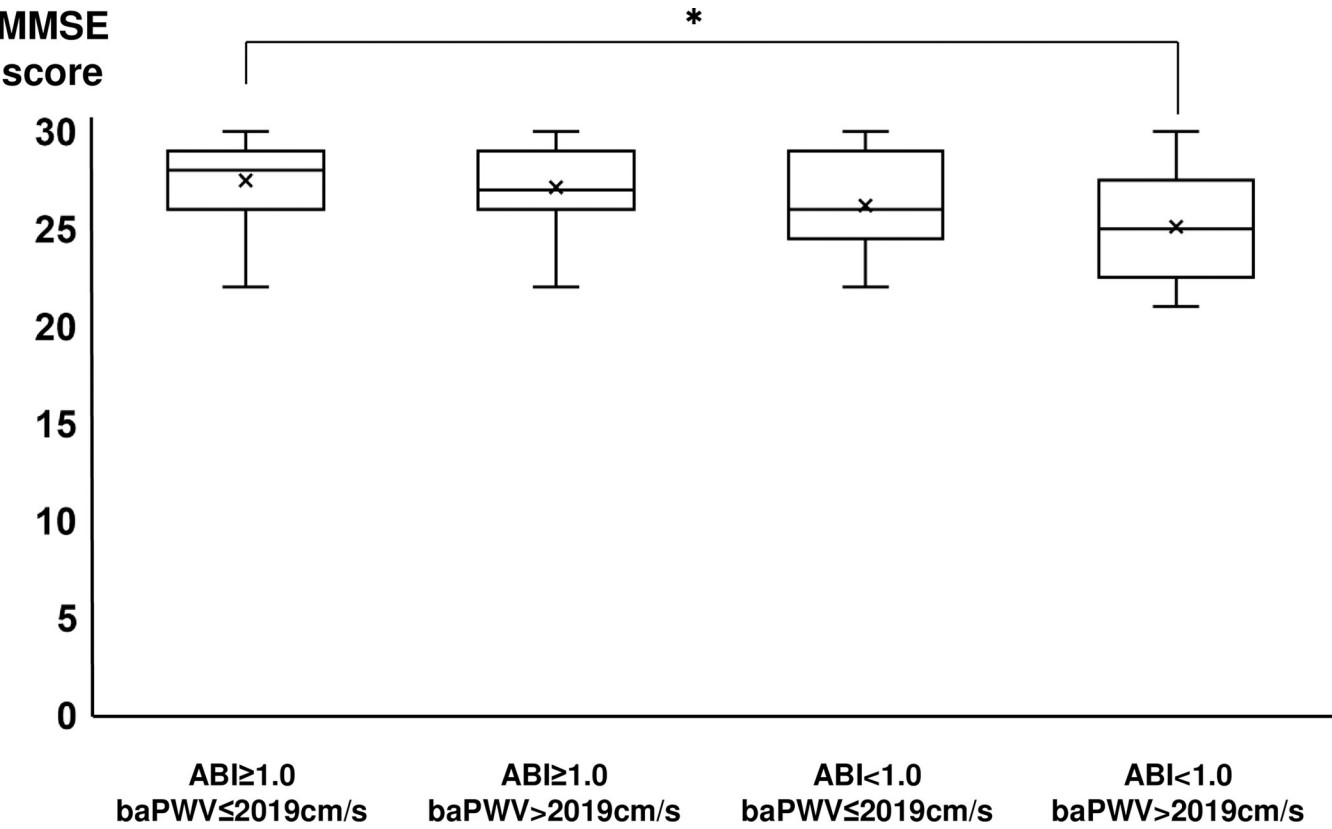

**Fig 2. Comparison of the MMSE scores among the four groups: ABI ≥ 1.0 and baPWV ≤ 2019 cm/s, ABI ≥ 1.0 and baPWV > 2019 cm/s, ABI < 1.0 and baPWV ≤ 2019 cm/s, and ABI < 1.0 and baPWV > 2019 cm/s.** The MMSE score significantly decreased with lower ABI and higher baPWV ($p < 0.05$). Bonferroni correction and Tukey's HSD tests revealed that the MMSE scores for the group with ABI < 1.0 and baPWV > 2019 cm/s were significantly lower than those for the group with ABI ≥ 1.0 and baPWV ≤ 2019 cm/s. MMSE, Mini-Mental State Examination; ABI, ankle-brachial pressure index; baPWV, brachial-ankle pulse wave velocity.

cognitive decline. The MMSE score decreased with lower ABI and higher baPWV; however, a higher baPWV was not significantly associated with the MMSE score. The sub-scores for orientation and immediate recall were significantly lower in the lower ABI group than in the higher ABI group.

Peripheral artery disease due to a lower ABI can be highly complicated by stroke. In the Reduction of Atherothrombosis for Continued Health Registry, 8.5% of patients with prior stroke and a transient ischemic attack had peripheral artery disease and 23.0% of patients with peripheral artery disease had stroke and a transient ischemic attack [25]. Moreover, a lower ABI predicts poor 3-month outcomes in patients with stroke [26]. There have been several population-based studies on the relationship between ABI and cognitive function [3, 9, 10]. It is reported that peripheral arterial disease was associated with a lower score for mental status [27]. A large community-based study reported an association of lower ABI with a decline in cognitive function over 7 years of follow-up [9]. Moreover, another study with community-dwelling older individuals reported that a lower ABI was an independent risk factor for cognitive decline [11]. In this study, we excluded patients with CMBs and severe WMLs on MRI, which are hallmark findings of cerebral small vessel disease (cSVD), which has been associated with cognitive decline [24, 28–30]. Our findings indicate that regardless of cSVD hallmarks other than lacunar infarction, a lower ABI was associated with cognitive decline.

**Table 2. Associations between multiple factors and decrease in MMSE scores.**

| | Univariate analysis | | Multivariate analysis | |
|---|---|---|---|---|
| | Predictive value | p value | Predictive value | p value |
| Age | -0.039 | 0.013 | -0.019 | 0.331 |
| Sex (female) | 0.044 | 0.814 | | |
| Body mass index | 0.121 | 0.011 | 0.099 | 0.039* |
| Education | 0.139 | 0.057 | 0.031 | 0.724 |
| Hypertension | -0.020 | 0.917 | | |
| Diabetes mellitus | -0.129 | 0.536 | | |
| Dyslipidemia | 0.054 | 0.768 | | |
| Chronic kidney disease | -0.472 | 0.021 | -0.365 | 0.079 |
| Current smoker | -0.172 | 0.358 | | |
| Habitual drinker | -0.012 | 0.949 | | |
| Antihypertensive drug | -0.038 | 0.838 | | |
| Antidiabetic drug | -0.022 | 0.926 | | |
| NIHSS score | -0.198 | 0.108 | | |
| Location of infarction | | | | |
| Side of the lesion (left) | -0.138 | 0.444 | | |
| Corona radiata | 0.198 | 0.316 | | |
| Basal ganglia | -0.172 | 0.658 | | |
| Capsulae internae | -0.113 | 0.605 | | |
| Thalamus | 0.150 | 0.452 | | |
| Brain stem | -0.344 | 0.174 | | |
| MRI findings | | | | |
| DSWMH | -0.424 | 0.182 | | |
| PVH | -0.461 | 0.060 | -0.160 | 0.532 |
| Ankle brachial pressure index <1.0 | -0.808 | 0.005 | -0.731 | 0.015* |
| Brachial-ankle pulse wave velocity >2019 cm/s | -0.205 | 0.255 | | |

MMSE, Mini-Mental Scale Examination; NIHSS, National Institutes of Health Stroke Scale; MRI, magnetic resonance imaging; DSWMH, deep and subcortical white matter hyperintensity; PVH, periventricular hyperintensity.

* indicates <0.05.

Contrastingly, some studies have reported an association between the MMSE scores and PWV. This inconsistency could be attributed to differences in the population targets. Several studies have reported an association of cognitive decline with arterial stiffness. Moreover,

**Table 3. Comparison of MMSE sub-scores.**

| | Ankle-brachial pressure index <1.0 | Ankle-brachial pressure index ≥1.0 |
|---|---|---|
| Orientation | 10 (9–10) | 10 (10–10) |
| Immediate recall | 3 (3–3) | 3 (3–3) |
| Attention and calculation | 2 (1–5) | 5 (2–5) |
| Delayed recall | 2 (2–3) | 2 (2–3) |
| Language | 9 (8–9) | 9 (9–9) |
| Visuospatial cognition | 1 (1–1) | 1 (1–1) |

MMSE, Mini-Mental State Examination. Data are presented as median (25% interquartile range [IQR] to 75% IQR). The sub-scores for orientation and immediate recall were significantly lower in the ABI < 1.0 group after adjustment for age, body mass index, education, and chronic kidney disease (p = 0.012 and 0.011, respectively).

numerous studies have reported an association of cfPWV with cognitive decline. The baPWV is measured between two sites along the arterial system and is preferred as it is easy to perform; however, cfPWV has been established as a more robust measure and is the gold standard. Studies using a modest sample of elderly individuals have suggested that arterial stiffness may contribute to the overlap between cSVD and amyloid β deposition in the brain [31, 32]. The baPWV, which is indicative of arterial stiffness, could reflect cSVD. The cSVD is associated with cognitive decline; however, we did not observe a significant association of a higher baPWV with the MMSE score. Similarly, in patients undergoing dialysis, compared with ABI, baPWV had better predictive power for mortality [33]. Additionally, among patients with acute stroke, compared with ABI, baPWV had a weaker predictive power for 3-month outcomes [26]. This may be explained by several factors. The baPWV was underestimated in patients with peripheral artery disease. The baPWV represents arterial stiffness, which increases owing to atherosclerosis. However, ABI is indicative of arterial stenosis or obstruction, which in turn indicates a progressed atherosclerosis stage. Therefore, ABI may indicate severe atherosclerosis, which has an impact on cognitive decline compared with baPWV.

In our study, the sub-scores for orientation and immediate recall, but not delayed recall, were lower in the ABI < 1.0 group than in the ABI ≥ 1.0 group. In addition, the sub-scores for attention and calculation were relatively low in all patients. Patients with mild cognitive impairment and vascular features tend to exhibit decreased frontal lobar function including self-motivation and executive function [34]. In this study, sub-score analyses did not strongly suggest memory function, but attention, self-motivation, and executive function, which was not consistent with cognitive decline and vascular features. These results suggest that cerebral microangiopathy might contribute to impaired cognitive function. The mechanisms underlying the association between the pathological and physiological mechanisms remain unclear. There is a need for further studies on the correlations between lower ABI and pathology to validate the specificity of the relationship with cognitive decline.

This study has several limitations. First, this study was a retrospective single center study. The selection bias was limitation. In addition, we divided the patients into four groups according to the ABI and baPWV; however, the sample size of these two groups was very small, potentially biasing some of the results. In this study, we excluded patients diagnosed with dementia, including strategic single-infarct dementia, before or after stroke onset. Moreover, we excluded patients with CMBs and severe WMLs. This study aimed to investigate the relationship between ABI/baPWV and cognitive function in patients excluding such factors. Ideally, we should investigate healthy subjects who underwent brain health screening tests. Second, cognitive assessment was only performed using the MMSE. Other tests, including the Montreal cognitive assessment, might detect slight cognitive decline.

## Conclusions

Among patients with lacunar infarction, there was an independent association of a lower ABI with cognitive decline. A higher PWV, which indicates peripheral arterial stiffness causing endothelial dysfunction, was not associated with cognitive decline. Patients with lacunar infarction who have small vessel damage might already have a decreased PWV, which might impede evaluation.

## Supporting information

**S1 Table. SBP and DBP of the four limbs and bilateral ankle-brachial pressure index and brachial-ankle pulse wave velocity.**
(DOCX)

**S2 Table. Characteristics of each group.**
(DOCX)

**S3 Table. Background characteristics of the patients, including the patients with cerebral microbleeds and severe white matter lesions.**
(DOCX)

**S4 Table. Associations between multiple factors and decrease in MMSE scores in all patients, including the patients with cerebral microbleeds and severe white matter lesions (n = 268).**
(DOCX)

**S1 Data. All relevant data of the study.**
(XLSX)

# Acknowledgments

We would like to sincerely thank Ms. Akiko Hirata, Ms. Masami Nishino, Ms. Masako Fukuta, and Ms. Kanami Ogawa for their technical assistance.

# Author Contributions

**Conceptualization:** Masahiro Nakamori, Hayato Matsushima, Eiji Imamura, Tatsuya Mizoue.

**Data curation:** Masahiro Nakamori, Hayato Matsushima, Keisuke Tachiyama, Yuki Hayashi.

**Formal analysis:** Masahiro Nakamori, Hayato Matsushima, Keisuke Tachiyama, Yuki Hayashi.

**Investigation:** Masahiro Nakamori, Hayato Matsushima, Keisuke Tachiyama, Yuki Hayashi.

**Methodology:** Masahiro Nakamori, Hayato Matsushima, Keisuke Tachiyama, Yuki Hayashi.

**Resources:** Eiji Imamura.

**Supervision:** Eiji Imamura, Tatsuya Mizoue, Shinichi Wakabayashi.

**Validation:** Eiji Imamura, Tatsuya Mizoue, Shinichi Wakabayashi.

**Visualization:** Eiji Imamura.

**Writing – original draft:** Masahiro Nakamori, Keisuke Tachiyama, Eiji Imamura, Tatsuya Mizoue, Shinichi Wakabayashi.

**Writing – review & editing:** Masahiro Nakamori, Tatsuya Mizoue, Shinichi Wakabayashi.

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
