## [Decision Letter · Decision Letter 0]

15 Sep 2021

PONE-D-21-22076Association of ankle-brachial index with cognitive decline in patients with lacunar infarctionPLOS ONE

Dear Dr. Nakamori,

Thank you for submitting your manuscript to PLOS ONE. After careful consideration, we feel that it has merit but does not fully meet PLOS ONE’s publication criteria as it currently stands. Therefore, we invite you to submit a revised version of the manuscript that addresses the points raised during the review process.

Two Reviewers well assessed this manuscript.  However, several major revisions are needed in the present form.  See the Reviewers’ comments and respond them appropriately.

We look forward to receiving your revised manuscript.

Kind regards,

Masaki Mogi

Academic Editor

PLOS ONE

Journal Requirements:

Reviewers' comments:

Reviewer's Responses to Questions

**Comments to the Author**

1. Is the manuscript technically sound, and do the data support the conclusions?

Reviewer #1: Partly

Reviewer #2: Yes

2. Has the statistical analysis been performed appropriately and rigorously? 

Reviewer #1: No

Reviewer #2: Yes

3. Have the authors made all data underlying the findings in their manuscript fully available?

Reviewer #1: Yes

Reviewer #2: Yes

4. Is the manuscript presented in an intelligible fashion and written in standard English?

Reviewer #1: Yes

Reviewer #2: No

5. Review Comments to the Author

Reviewer #1: This manuscript looks at the relationship between arterial health and cognition in a group of patients with lacunar infarct, but no indication of small vessel disease. The authors found an association between cognition ankle-brachial pressure index (ABI), but not brachial-ankle pulse wave velocity (baPWV). This manuscript is interesting and clear, though there are some important issues. The biggest issue with this study is that since there is no control group, it’s difficult to interpret the results in terms of stroke vs other metrics. Stroke and lacune severity were not used in the model and it is possible that these are co-linear with for example ABI. Also, the choices of what to include or not in terms of recruitment criteria seems somewhat arbitrary, further eroding the reader’s understanding of what the authors are trying to achieve. A better justification of the design would be helpful to the reader. Major and minor comments are detailed below:

Major:

The lack of a control groups makes it difficult to disentangle some of the variables at play. For example, since all participants have a stroke, it may be that stroke severity is related to ABI for example. The addition of a control group or a more thorough list of co-variates would enhance the interpretability of the results.

Intro: The authors could do a more convincing job at explaining why it is important to look at the relationship between artery health and cognition in stroke patients with lacunas, but in the absence of WMH, microbleeds and medial temporal atrophy. In the current writing, it basically sounds like this is interesting because that’s what we did. Also, it is not clear in the concluding sentence of the intro that participants are stroke patients with lacunes. Why include lacunes specifically?

No correction for multiple comparison was performed so some of the results may be spurious

The authors used the median PWV of their cohort to divide the group. This may not be a meaningful cutoff in terms of physiology. It might make more sense to either do a linear regression rather than dichotomize their variable (especially since this has been shown to lead to spurious effects). Alternately, using a value from the literature shown to be associated with cognitive effects may make more sense. These limitations in the interpretation of a lack of correlation for PWV should be acknowledged in the discussion.

More information on the multivariate analysis would help evaluate the manuscript. Did the authors mean least squares? How were the factors entered? All at once? Sequentially?

The authors have split the MMSE into different scores, but there is no discussion of the significance of different domains being or not decreased with ABI, etc. Discussion of these results should be added.

Minor:

Table 1. The two last rows are missing units

Figure 2. Some indication inside the figure (star perhaps) of which combinations were found to be significantly different would be helpful.

Reviewer #2: In the manuscript, Nakamori et al. have shown the “Association of ankle-brachial index with cognitive decline in patients with lacunar infarction". There were several flaws.

Major comments

Abstract

#1: The number of the patients with ABI<1.0

Please provide the number of the patients with ABI<1.0

#2: The IQR of MMSE score

It is helpful if the IQR of MMSE score was provided.

Methods

#2: The power analysis

Did the authors perform the power analysis? If not, there was some possibility that the results were derived by chance.

#3: Validation data for the device used for measuring ABI

Is there any validation paper for the device used for measuring ABI? If there were, please provide those.

#4: ASO patients

Were there any patients with symptomatic ASO. Were there any patients who needs to be performed PTA?

#5: Exclusion of the patients with dementia

The each IQR of MMSE was 26 or 29. There might be some patients with dementia in the groups of the lowest quartile of MMSE score.

#6: Exclusion of patients with one or more microbleeds, severe WMLs, or severe medial temporal atrophy

The authors excluded the patients with one or more microbleeds, severe WMLs, or severe medial temporal atrophy. However, the silent lacunae was not taken into account. Some of the patients with multiple silent lacunaes were associated with cognitive impairment, and might be confounder in this study results.

#4: The level of BP

It would be helpful if the BP levels were provided. I have missed the data about BP.

#5: Reference for MMSE

Please provide the reference for MMSE score

Results

#7: Laterality/ location of stroke

Was there any difference in the cognitive function according to lacuna stroke lesion location and laterality?

#8: Treatment of cardiovascular risk factors

Cardio-cerebro vascular risk factors such as hypertension and T2DM would be associated with cognitive function. Did the antihypertensive treatment or hypoglycemic agents were investigated in the analysis?

#9: Antihypertensive medication before and during in-hospital and after discharge.

In this study, all the patients underwent the MMSE and ABI/baPWV measurements within three days of admission. This might be similar to the question to #3. Are there any antihypertensive medication before and during in-hospital? Such as intravenous calcium channel blockade might be associated with blood pressure on the acute phase. And this might be associated with the study results.

#10: Left and right ABI

Were there any difference in the laterality of ABI for the MMSE score?

6. PLOS authors have the option to publish the peer review history of their article (what does this mean?). If published, this will include your full peer review and any attached files.

Reviewer #1: No

Reviewer #2: **Yes: **Michiaki Nagai

---

## [Author Response · Author response to Decision Letter 0]

20 Oct 2021

Thank you for reviewing our manuscript. We appreciate your comments and suggestions. We have revised the manuscript accordingly. Our point-by-point responses are presented below.

Response to the Editor

Comment 1: Please ensure that your manuscript meets PLOS ONE's style requirements, including those for file naming. The PLOS ONE style templates can be found at　https://journals.plos.org/plosone/s/file?id=wjVg/PLOSOne_formatting_sample_main_body.pdf and　https://journals.plos.org/plosone/s/file?id=ba62/PLOSOne_formatting_sample_title_authors_affiliations.pdf

Response 1: Thank you for the comment. We have ensured that the manuscript meets the style requirements of PLOS ONE.

Comment 2: Thank you for stating the following financial disclosure: 

Response 2: We have added the following sentence in the Funding section and the cover letter:

The authors received no specific funding for this work.

Comment 3: We note that you have stated that you will provide repository information for your data at acceptance. Should your manuscript be accepted for publication, we will hold it until you provide the relevant accession numbers or DOIs necessary to access your data. If you wish to make changes to your Data Availability statement, please describe these changes in your cover letter and we will update your Data Availability statement to reflect the information you provide.

Response 3: We do not intend to make any changes in our Data Availability statement.

 

Response to the reviewers

Reviewer #1: This manuscript looks at the relationship between arterial health and cognition in a group of patients with lacunar infarct, but no indication of small vessel disease. The authors found an association between cognition ankle-brachial pressure index (ABI), but not brachial-ankle pulse wave velocity (baPWV). This manuscript is interesting and clear, though there are some important issues. The biggest issue with this study is that since there is no control group, it’s difficult to interpret the results in terms of stroke vs other metrics. Stroke and lacune severity were not used in the model and it is possible that these are co-linear with for example ABI. Also, the choices of what to include or not in terms of recruitment criteria seems somewhat arbitrary, further eroding the reader’s understanding of what the authors are trying to achieve. A better justification of the design would be helpful to the reader. Major and minor comments are detailed below:

Response: We appreciate your insightful comments and advice. According to your advice, we have provided point-by-point responses to each of your comments below and denoted the corresponding revisions in the manuscript in red text.

Comment 1: The lack of a control groups makes it difficult to disentangle some of the variables at play. For example, since all participants have a stroke, it may be that stroke severity is related to ABI for example. The addition of a control group or a more thorough list of co-variates would enhance the interpretability of the results.

Response 1: We appreciate your advice. Unfortunately, owing to the study design, obtaining a purely normal control group was difficult. However, we have added the National Institutes of Health Stroke Scale (NIHSS) scores (stroke severity scores) as baseline data, which were statistically analyzed. We found no significant association between the NIHSS and Mini-Mental State Examination (MMSE) scores. In addition, we found no association between NIHSS scores and ankle-brachial pressure index/brachial-ankle pulse wave velocity (ABI/baPWV). In this study, we focused on lacunar infarction (mild stroke). Lacunar stroke itself does not directly affect cognitive function, because it does not damage the cerebral cortex. Therefore, we considered the presence of stroke lesions and analyzed their effect on cognitive function. However, we did not find any association. Since we focused on patients with lacunar infarction, the neurological symptoms were mild. Some patients had almost no symptoms. We considered patients with NIHSS score 0 as almost normal controls and analyzed the effects, which revealed no significant relationship with MMSE score, ABI, and baPWV. Moreover, we have added a list of covariates to enhance the statistical rigorousness. We have revised the tables and added the description in the manuscript as follows:

Lines 130-134

We recorded baseline clinical characteristics, including age, sex, body mass index (BMI), duration of education, complications (hypertension, diabetes mellitus, dyslipidemia, and chronic kidney disease), current smoking, habitual drinking, and medication before admission (antihypertensive and antidiabetic drugs). The severity of stroke was evaluated using the National Institutes of Health Stroke Scale (NIHSS) [23]. 

23. Lyden P, Brott T, Tilley B, Welch KM, Mascha EJ, Levine S, et al. Improved reliability of the NIH Stroke Scale using video training. NINDS TPA Stroke Study Group. Stroke. 1994;25: 2220-2226.

Lines 177-183

We evaluated the influence of stroke severity on cognitive function or ABI/baPWV. Linear regression analyses showed no association between NIHSS and MMSE scores (p=0.108). Further, there was also no association between NIHSS scores and ABI or baPWV (p=0.376, p=0.233, respectively). We considered patients with NIHSS score 0 as almost normal controls and analyzed the effects, which revealed no significant relationship with MMSE score, ABI, or baPWV (p=0.646, p=0.546, p=0.132, respectively).

Comment 2: Intro: The authors could do a more convincing job at explaining why it is important to look at the relationship between artery health and cognition in stroke patients with lacunas, but in the absence of WMH, microbleeds and medial temporal atrophy. In the current writing, it basically sounds like this is interesting because that’s what we did. Also, it is not clear in the concluding sentence of the intro that participants are stroke patients with lacunes. Why include lacunes specifically?

Response 2: We appreciate the insightful comments. Regarding the recruitment criteria, we had to exclude cases of other pathologies, primarily those of Alzheimer’s pathology. Medial temporal atrophy is strongly associated with Alzheimer’s pathology. Microbleeds, especially cortical microbleeds are associated with amyloid (Alzheimer’s) pathology. In addition, severe white matter lesions are sometimes associated with neurodegenerative diseases such as leukoencepalopathy. We had to evaluate the interaction with covariates statistically; however, other neurodegenerative diseases or pathologies had to be excluded. Therefore, we decided the inclusion/exclusion criteria. We have explained the reason in detail in the revised manuscript. 

Moreover, we evaluated stroke patients focusing on lacunar infarction, because the basic mechanism of stroke differs among stroke subtypes. This difference in the mechanism could affect the results of this study. Additionally, in other stroke types, i.e., other than lacunar infarction, the stroke lesions usually involve the cerebral cortex, which itself can affect the cognitive function. Therefore, we focused on patients with lacunar infarction.

We have revised the Introduction section as follows:

Lines 66-80

We aimed to investigate the relationship of ABI and baPWV with cognitive function. In this study, we focused on the first-ever acute stroke and a diagnosis of lacunar infarction because the basic mechanism of stroke differs among stroke subtypes. Additionally, in other stroke types, i.e., other than lacunar infarction, the stroke lesions usually involve the cerebral cortex, which itself can affect the cognitive function. In addition, we had to exclude cases of neurodegenerative diseases, primarily those of Alzheimer’s pathology. Therefore, we tried to exclud cases with cerebral microbleeds (CMBs), white matter lesions (WMLs), and medial temporal atrophy. CMBs, especially lobar type CMBs, are associated with amyloid pathology. Severe WMLs are sometimes associated with neurodegenerative diseases such as leukoencepalopathy. Medial temporal atrophy is strongly associated with Alzheimer’s pathology. Few reports have assessed the association between atherosclerosis and cognitive decline by excluding such factors. In this study, we investigated the relationship of ABI and baPWV with cognitive function in patients with first-ever lacunar infarction and without CMBs, WMLs, and medial temporal atrophy.

Comment 3: No correction for multiple comparison was performed so some of the results may be spurious

Response 3: We appreciate your comment. Accordingly, for multiple comparisons, we have analyzed the data using one-way analysis of variance and Tukey’s honestly significant difference test. We have revised the manuscript (Methods and Results section) and Figure 2.

Lines 143-149

Univariate analysis was used to investigate the association of MMSE scores with several factors, and a value of p = 0.10 was used to indicate statistical significance. Subsequently, multivariate analysis was performed with selected factors determined from univariate analysis. In multivariate analysis, the least squares test was performed with the selected factors, which were entered simultaneously. For multiple comparisons, the data were analyzed using one-way analysis of variance (ANOVA) and Tukey’s honestly significant difference (HSD) test.

Lines 194-202

The median baPWV was 2019 cm/s. The patients were categorized into the following four groups: patients with ABI ≥ 1.0 and baPWV ≤ 2019 cm/s; ABI ≥ 1.0 and baPWV > 2019 cm/s; ABI < 1.0 and baPWV ≤ 2019 cm/s; or ABI < 1.0 and baPWV > 2019 cm/s. S2 Table presents the characteristics of each group. Fig 2 shows the MMSE scores for each group. ANOVA showed that the MMSE scores were directly and inversely correlated with ABI and baPWV, respectively (p < 0.05). Tukey’s HSD tests revealed that the MMSE scores for the group with ABI < 1.0 and baPWV > 2019 cm/s were significantly lower than those for the group with ABI ≥ 1.0 and baPWV ≤ 2019 cm/s.

Comment 4: The authors used the median PWV of their cohort to divide the group. This may not be a meaningful cutoff in terms of physiology. It might make more sense to either do a linear regression rather than dichotomize their variable (especially since this has been shown to lead to spurious effects). Alternately, using a value from the literature shown to be associated with cognitive effects may make more sense. These limitations in the interpretation of a lack of correlation for PWV should be acknowledged in the discussion.

Response 4: We appreciate the insightful comments and advice. Regarding baPWV, we performed linear regression as well as dichotomized analysis. The meaningful cutoff values of baPWV have varied among studies. Thus, we used the median value in this study. 

Lines 127-129

For baPWV, we used the higher ankle side and the cases were categorized into two groups based on the median value of the participants. In addition, we performed linear analysis for baPWV.

Lines 218-222

We also performed linear analysis for baPWV. Univariate analysis showed that baPWV was associated with MMSE scores (p=0.011). However, multivariate analysis using age, BMI, education, and chronic kidney disease as covariates showed that baPWV was not significantly associated with MMSE scores (p=0.117).

Comment 5: More information on the multivariate analysis would help evaluate the manuscript. Did the authors mean least squares? How were the factors entered? All at once? Sequentially?

Response 5: We appreciate your valuable questions and comments. For multivariate analysis, we selected factors using univariate analysis, in which a value of p = 0.10 was used to indicate statistical significance. Then, the least squares test was performed and all factors were entered simultaneously. We have explained the process of multivariate analysis in detail as follows:

Lines 143-147

Univariate analysis was used to investigate the association of MMSE scores with several factors, and a value of p = 0.10 was used to indicate statistical significance. Subsequently, multivariate analysis was performed with selected factors determined from univariate analysis. In multivariate analysis, the least squares test was performed with the selected factors, which were entered simultaneously.

Comment 6: The authors have split the MMSE into different scores, but there is no discussion of the significance of different domains being or not decreased with ABI, etc. Discussion of these results should be added.

Response 6: We appreciate your advice. We have added the information about the domains of MMSE in the Discussion section as follows:

Lines 282-293

In our study, the sub-scores for orientation and immediate recall, but not delayed recall, were lower in the ABI < 1.0 group than in the ABI ≥ 1.0 group. In addition, the sub-scores for attention and calculation were relatively low in all patients. Patients with mild cognitive impairment and vascular features tend to exhibit decreased frontal lobar function including self-motivation and executive function [35]. In this study, sub-score analyses did not strongly suggest memory function, but attention, self-motivation, and executive function, which was not consistent with cognitive decline and vascular features. These results suggest that cerebral microangiopathy might contribute to impaired cognitive function. The mechanisms underlying the association between the pathological and physiological mechanisms remain unclear. There is a need for further studies on the correlations between lower ABI and pathology to validate the specificity of the relationship with cognitive decline.

35. Frisoni GB, Galluzzi S, Bresciani L, Zanetti O, Geroldi C Mild cognitive 

impairment with subcortical vascular features: clinical characteristics and outcome. J 

Neurol. 2002;249: 1423-1432.

Comment 7: Table 1. The two last rows are missing units

Response 7: We apologize for the error. We have revised the Table.

Comment 8: Figure 2. Some indication inside the figure (star perhaps) of which combinations were found to be significantly different would be helpful.

Response 8: We appreciate your advice. We have revised the figure and provided an indication (star) inside the figure for ease of understanding.

Reviewer #2: In the manuscript, Nakamori et al. have shown the “Association of ankle-brachial index with cognitive decline in patients with lacunar infarction". There were several flaws.

Response: We appreciate your insightful comments and advice. We have provided point-by-point responses to each of your comments below and denoted the corresponding revisions in the manuscript in red text.

Abstract:

Comment 1: The number of the patients with ABI<1.0

Please provide the number of the patients with ABI<1.0

Response 1: We provided the number of the patients with ABI<1.0 in Abstract.

Line 30

We analyzed 176 patients with stroke (age 72.5 ± 11.4 years, 67 females). The median score of the Mini-Mental State Examination (MMSE) was 27. The number of the patients with ABI<1.0 was 19 (10.8%).

Comment 2: The IQR of MMSE score

It is helpful if the IQR of MMSE score was provided.

Response 2: We have mentioned the IQR of MMSE scores in the manuscript and Table 1.

Methods

Comment 3: The power analysis

Did the authors perform the power analysis? If not, there was some possibility that the results were derived by chance.

Response 3: We appreciate your valuable comments. We have performed power analysis in advance. We have added the relevant information in the manuscript as follows:

Lines 140-143

We calculated the required sample size for this study according to previous studies that compared MMSE scores with ABI/baPWV or CMBs [11, 25]. Based on an alpha level of 0.05 and power of 0.80, we estimated that we would require at least 128 participants.

11. Sugawara N, Yasui-Furukori N, Umeda T, Kaneda A, Sato Y, Takahashi I, et al. 

Comparison of ankle-brachial pressure index and pulse wave velocity as markers of 

cognitive function in a community-dwelling population. BMC Psychiatry. 2010;10: 46.

25. Nakamori M, Hosomi N, Tachiyama K, Kamimura T, Matsushima H, Hayashi Y, et 

al. Lobar microbleeds are associated with cognitive impairment in patients with lacunar 

infarction. Sci Rep 2020;10: 16410.

Comment 4: Validation data for the device used for measuring ABI

Is there any validation paper for the device used for measuring ABI? If there were, please provide those.

Response 4: We have added the study that has validated the method of ABI/baPWV measurement as a reference.

Lines 121-125

MMSE scores were recorded and ABI/baPWV measurements were performed within three days of admission for all patients [21]. The accuracy of the method of ABI/baPWV measurement has been validated previously [22]. ABI/baPWV measurements were performed using BP-203RPE III (OMRON HEALTHCARE Co., Ltd., Kyoto, Japan).

21. Folstein MF, Folstein SE, McHugh PR. "Mini-mental state". A practical 

method for grading the cognitive state of patients for the clinician. J Psychiatr Res. 

1975;12: 189-198.

22. Yamashina A, Tomiyama H, Takeda K, Tsuda H, Arai T, Hirose K, et al. Validity, 

Reproducibility, and clinical significance of noninvasive brachial-ankle pulse wave 

velocity measurement. Hypertens Res. 2002;25: 359-364.

Comment 5: ASO patients

Were there any patients with symptomatic ASO. Were there any patients who needs to be performed PTA?

Response 5: In this study, there were no patients who had symptomatic ASO and in whom PTA had to be performed.

Lines 174-183

Data pertaining to systolic and diastolic blood pressure in the four limbs and bilateral ABI and baPWV are shown in S1 Table. We investigated the association between the laterality of ABI and the MMSE score but found no association. We evaluated the influence of stroke severity on cognitive function or ABI/baPWV. Linear regression analyses showed no association between NIHSS and MMSE scores (p=0.108). Further, there was also no association between NIHSS scores and ABI or baPWV (p=0.376, p=0.233, respectively). We considered patients with NIHSS score 0 as almost normal controls and analyzed the effects, which revealed no significant relationship with MMSE score, ABI, or baPWV (p=0.646, p=0.546, p=0.132, respectively).

Comment 6: Exclusion of the patients with dementia

The each IQR of MMSE was 26 or 29. There might be some patients with dementia in the groups of the lowest quartile of MMSE score.

Response 6: The minimum MMSE score of patients in this study was 21. However, we did not diagnose dementia on the bases of lower scores on cognitive tests. We diagnosed dementia according to the clinical history (education, occupation), data (blood, imaging, neurological, and cognitive tests), and interview with patients and their families. In this study, despite lower MMSE scores, we did not diagnose these patients with dementia. 

Comment 7: Exclusion of patients with one or more microbleeds, severe WMLs, or severe medial temporal atrophy.

The authors excluded the patients with one or more microbleeds, severe WMLs, or severe medial temporal atrophy. However, the silent lacunae was not taken into account. Some of the patients with multiple silent lacunaes were associated with cognitive impairment, and might be confounder in this study results.

Response 7: We appreciate your insightful comments. In this study, we included patients with first-ever stroke. As you mentioned, it is important to consider the presence of silent lacunar lesions. However, it is difficult to definitely differentiate lacunar lesions from other lesions such as dilation of the perivascular space and white matter lesions. In fact, inter-rater match was poor. Thus, it was impossible to perform quantitative analysis. In addition, there is no gold standard grading for lacunar burden. However, according to a previous report, the number of lacunae was graded as follows: grade 0, absent; grade 1, 1 to 2 lacunae; grade 2, 3 to 5 lacunae; and grade 3, 6 lacunae. In this study, two stroke neurologists analyzed the patients, and all patients were determined as grade 3 or less. This was because we excluded patients with severe white matter lesions. We believe that we excluded the confounder using this method. We have revised the text as follows:

Lines 107-119

The severity of WMLs (deep and subcortical white matter hyperintensity [DSWMH] and periventricular hyperintensity [PVH]) was rated visually on fluid-attenuated inversion recovery images using the Fazekas scale (DSWMH: grade 1, punctuate; grade 2, early confluence; and grade 3, confluent; and PVH: grade 1, caps or lining; grade 2, bands; and grade 3, irregular extension into the deep white matter) [18]. Patients with WMLs (DSWMH or PVH) of grades 3 were assigned to the severe WML groups. The degree of medial temporal atrophy was semi quantitatively evaluated as described in previous reports [19], using a 5-point score scale ranging from 0 (no atrophy) to 4 (severe atrophy). In addition, according to a previous report, we evaluated the presence of silent lacunar lesions and graded the number of lacunae as follows: grade 0, absent; grade 1, 1 to 2 lacunae; grade 2, 3 to 5 lacunae; and grade 3, 6 or more lacunae [20]. Two stroke neurologists (MN and KT) graded the patients after consensus.

20. Yamamoto Y, Akiguchi I, Oiwa K, Hayashi M, Kasai T, Ozasa K. Twenty-four-

hour blood pressure and MRI as predictive factors for different outcomes in patients 

with lacunar infarct. Stroke. 2002;33: 297-305.

Lines 160-166

A total of 826 patients were diagnosed with lacunar infarction; of these, 468 patients were admitted for their first-ever stroke. We excluded 43 patients without MRI data, 25 patients without MMSE scores, and 127 patients diagnosed with dementia, including strategic single-infarct dementia (n = 3), before or after stroke onset. Moreover, we excluded 97 patients with CMBs, severe WMLs, or severe medial temporal atrophy. Ultimately, we analyzed 176 patients with stroke (age: 72.5 ± 11.4 years, 67 females; Fig 1). Regarding to the silent lacunar lesion, all patients were judged as grade 3 or less.

Comment 8: The level of BP

It would be helpful if the BP levels were provided. I have missed the data about BP.

Response 8: We have added the information about blood pressure in S1 Table.

Lines 173-176

Table 1 shows the patient background characteristics. Data pertaining to systolic and diastolic blood pressure in the four limbs and bilateral ABI and baPWV are shown in S1 Table.

Comment 9: Reference for MMSE

Please provide the reference for MMSE score.

Response 9: We have added the reference as follows:

Lines 121-125

MMSE scores were recorded and ABI/baPWV measurements were performed within three days of admission for all patients [21]. The accuracy of the method of ABI/baPWV measurement has been validated previously [22]. ABI/baPWV measurements were performed using BP-203RPE III (OMRON HEALTHCARE Co., Ltd., Kyoto, Japan).

21. Folstein MF, Folstein SE, McHugh PR. "Mini-mental state". A practical 

method for grading the cognitive state of patients for the clinician. J Psychiatr Res. 

1975;12: 189-198.

22. Yamashina A, Tomiyama H, Takeda K, Tsuda H, Arai T, Hirose K, et al. Validity, 

reproducibility, and clinical significance of noninvasive brachial-ankle pulse wave 

velocity measurement. Hypertens Res. 2002;25: 359-364.

Results

Comment 10: Laterality/ location of stroke

Was there any difference in the cognitive function according to lacuna stroke lesion location and laterality?

Response 10: We have added the laterality and location of stroke lesions in baseline data and performed statistical analysis. We did not find any significant association. We have revised the Tables.

Comment 11: Treatment of cardiovascular risk factors

Cardio-cerebro vascular risk factors such as hypertension and T2DM would be associated with cognitive function. Did the antihypertensive treatment or hypoglycemic agents were investigated in the analysis?

Response 11: We have added the number of patients with previous antihypertensive or antidiabetic treatment in the Tables and performed the statistical analysis. We did not find any significant association.

Lines 130-133

We recorded baseline clinical characteristics, including age, sex, body mass index (BMI), duration of education, complications (hypertension, diabetes mellitus, dyslipidemia, and chronic kidney disease), current smoking, habitual drinking, and medication before admission (antihypertensive and antidiabetic drugs).

Comment 12: Antihypertensive medication before and during in-hospital and after discharge.

In this study, all the patients underwent the MMSE and ABI/baPWV measurements within three days of admission. This might be similar to the question to ＃11. Are there any antihypertensive medication before and during in-hospital? Such as intravenous calcium channel blockade might be associated with blood pressure on the acute phase. And this might be associated with the study results.

Response 12: 

In this study, we focused on patients with acute ischemic stroke in whom the systolic blood pressure was maintained under 220 mmHg. In the hospital, no patient required antihypertensive medications such as intravenous calcium channel blockers at the time of parameter measurement within three days of admission. Because some patients moved to other clinics or rehabilitation hospitals, we could not get the information about the use of antihypertensive medication after discharge. We have added the following information in the manuscript:

Lines 166-171

In this study, no patient had symptomatic peripheral artery disease and required percutaneous transluminal angioplasty. The systolic blood pressure in all patients was maintained under 220 mmHg. No patient required antihypertensive medications such as intravenous calcium channel blockers at the time of parameter measurement within three days of admission.

Comment 13: Left and right ABI

Were there any difference in the laterality of ABI for the MMSE score?

Response 13: 

We investigated the association between the laterality of ABI and the MMSE scores but did not find any association. We have added this information in the manuscript.

Lines 174-177

Data pertaining to systolic and diastolic blood pressure in the four limbs and bilateral ABI and baPWV are shown in S1 Table. We investigated the association between the laterality of ABI and the MMSE scores, but found no association.

---

## [Decision Letter · Decision Letter 1]

17 Nov 2021

PONE-D-21-22076R1Association of ankle-brachial index with cognitive decline in patients with lacunar infarctionPLOS ONE

Dear Dr. Nakamori,

Thank you for submitting your manuscript to PLOS ONE. After careful consideration, we feel that it has merit but does not fully meet PLOS ONE’s publication criteria as it currently stands. Therefore, we invite you to submit a revised version of the manuscript that addresses the points raised during the review process.

Major revisions are needed in the present form. See the Reviewers' comments carefully and respond them appropriately.

We look forward to receiving your revised manuscript.

Kind regards,

Masaki Mogi

Academic Editor

PLOS ONE

Reviewers' comments:

Reviewer's Responses to Questions

**Comments to the Author**

1. If the authors have adequately addressed your comments raised in a previous round of review and you feel that this manuscript is now acceptable for publication, you may indicate that here to bypass the “Comments to the Author” section, enter your conflict of interest statement in the “Confidential to Editor” section, and submit your "Accept" recommendation.

Reviewer #1: (No Response)

Reviewer #2: All comments have been addressed

2. Is the manuscript technically sound, and do the data support the conclusions?

Reviewer #1: Partly

Reviewer #2: Yes

3. Has the statistical analysis been performed appropriately and rigorously? 

Reviewer #1: Yes

Reviewer #2: Yes

4. Have the authors made all data underlying the findings in their manuscript fully available?

Reviewer #1: Yes

Reviewer #2: Yes

5. Is the manuscript presented in an intelligible fashion and written in standard English?

Reviewer #1: Yes

Reviewer #2: Yes

6. Review Comments to the Author

Reviewer #1: This manuscript explores the relationship between ABI and baPWV and cognition, as assessed by MMSE, in patients suffering from lacunar stroke. Patients with significant microbleeds and SVD were excluded from the analysis. Data show that ABI, but not baPWV is associated with lower overall cognition, orientation and immediate recall.

This is an interesting and well-written study investigating these relationships in a unique population. My main concern is that the numerous exclusion criteria resulted in a more limited sample size, especially for the low ABI groups, which were of most interest in this study. Major and minor concerns about this study are detailed below:

Major:

The authors used numerous exclusion criteria to define their group, resulting in a limited sample of low ABI patients. This is unfortunate as their main results are in this group, but could also be driven by a number of other variable of interest which differs in this group. The authors should therefore better justify their choice of population, since this population is also not representative of the general population (since they all have a stroke).

Field strength/MRI model was not taken into account in the analyses as a covariate of no interest. This should be done, especially if the distribution of patients scanned on each model is not equal amongst the different groups.

Was multiple comparison correction used for the analyses presented? Several comparisons are made and should therefore be corrected for.

Since other aspects of health could explain some of the results shown for the four sub-groups, the demographic table should be expanded to include characteristics of the overall sample as shown currently, as well as for the four subgroups, rather than including this information in the supplemental materials.

Was Fazekas score, or better, WMH volume used in any of the analyses. Since WMH is likely to be related to cognition, it would be pertinent to use this in analyses as it could independently explain some of the variance in MMSE score.

The sample size in two of the groups is very small, potentially biasing some of the results. This should be acknowledged in the discussion.

Minor:

Fezekas score for each group should be reported in the demographics table.

In the discussion, the authors mention that baPWV is preferable, but cfPWV has been established as a more robust measure and is the gold standard in the literature. It is also non-invasive. These caveats of baPWV should be acknowledged in the discussion.

Reviewer #2: The manuscript was substantially revised.

Minor comments

#1: “exclud”→”exclude”

In line 72, “…Alzheimer’s pathology. Therefore, we tried to exclud cases with cerebral microbleeds…” should be changed as “…Alzheimer’s pathology. Therefore, we tried to exclude cases with cerebral microbleeds…”?

#2: p=0.10

In the methods, the authors described that “a value of p = 0.10 was used to indicate statistical significance.” Is there any literature that could refer to defend the choice of p=0.10?

7. PLOS authors have the option to publish the peer review history of their article (what does this mean?). If published, this will include your full peer review and any attached files.

Reviewer #1: No

Reviewer #2: **Yes: **Michiaki Nagai

---

## [Author Response · Author response to Decision Letter 1]

14 Dec 2021

Thank you for reviewing our manuscript. We appreciate your comments and suggestions. We have revised the manuscript in accordance with your comments. Our point-by-point responses to the comments are presented below.

Response to the reviewers

Reviewer #1: This manuscript explores the relationship between ABI and baPWV and cognition, as assessed by MMSE, in patients suffering from lacunar stroke. Patients with significant microbleeds and SVD were excluded from the analysis. Data show that ABI, but not baPWV is associated with lower overall cognition, orientation and immediate recall.

This is an interesting and well-written study investigating these relationships in a unique population. My main concern is that the numerous exclusion criteria resulted in a more limited sample size, especially for the low ABI groups, which were of most interest in this study. Major and minor concerns about this study are detailed below:

Response: We appreciate your insightful comments and advice. To address the main concern raised by you, we reanalyzed after adding the patients who were excluded based on MRI findings. We have responded to each of your comments below and denoted the corresponding revisions in the manuscript in red color.

Comment 1: The authors used numerous exclusion criteria to define their group, resulting in a limited sample of low ABI patients. This is unfortunate as their main results are in this group, but could also be driven by a number of other variable of interest which differs in this group. The authors should therefore better justify their choice of population, since this population is also not representative of the general population (since they all have a stroke).

Response 1: We appreciate your insightful comment. We tried to include patients with lacunar infarction and exclude patients with other etiologies. In fact, some cerebral microbleeds are associated with amyloid pathology, and some white matter lesions are associated with neurodegenerative diseases. However, cerebral microbleeds and white matter lesions are associated with small vessel diseases such as lacunar infarction. I also understand the critical point that you mentioned. To justify patient selection and perform comprehensive analyses, we added and analyzed the patients with CMBs and severe white matter lesions (who were excluded based on MRI findings) as well. The analyses revealed that low ABI was independently associated with the MMSE score. We have revised the manuscript and added the new results in the Supplemental Tables.

Lines 243-251

Finally, we added the patients with CMBs and severe white matter lesions (who were excluded based on MRI findings) and reanalyzed again (n=268). S3 shows the background characteristics of these patients. The median baPWV was 2086.5 cm/s and we divided the patients based on the median baPWV. We investigated the association of the MMSE score with the factors listed in S3. Univariate analysis revealed that the MMSE score was associated with age, BMI, education, chronic kidney disease, cerebral microbleeds, PVH, and ABI < 1.0 (p < 0.10), but not with baPWV > 2086.5 cm/s. Multivariate analysis revealed an independent association of BMI (p = 0.009) and ABI < 1.0 (p = 0.019) with the MMSE score (S4).

Comment 2: Field strength/MRI model was not taken into account in the analyses as a covariate of no interest. This should be done, especially if the distribution of patients scanned on each model is not equal amongst the different groups.

Response 2: Owing to the appointments in our institution, we used 1.5T MRI from July 1, 2011, and 3.0T MRI from April 2015 to December 31, 2018. This was incidental, not intentional. The imaging protocols were the same for the two groups. We investigated the differences in background characteristics between the two groups and found no significant differences. We have added the following text:

Lines 101-103

The imaging protocols were the same for the two MRI groups. We investigated the differences in background characteristics between the two groups and found no significant differences.

Comment 3: Was multiple comparison correction used for the analyses presented? Several comparisons are made and should therefore be corrected for.

Response 3: For multiple comparisons, we performed Bonferroni correction and Tukey’s honestly significant difference (HSD) test. We have revised the manuscript (Methods and Results section).

Lines 146-148

For multiple comparisons, the data were analyzed using one-way analysis of variance (ANOVA), and Bonferroni correction and Tukey’s honestly significant difference (HSD) test were performed.

Lines 201-203

Bonferroni correction and the HSD test revealed that the MMSE scores for the group with ABI < 1.0 and baPWV > 2019 cm/s were significantly lower than those for the group with ABI ≥ 1.0 and baPWV ≤ 2019 cm/s.

Comment 4: Since other aspects of health could explain some of the results shown for the four sub-groups, the demographic table should be expanded to include characteristics of the overall sample as shown currently, as well as for the four subgroups, rather than including this information in the supplemental materials.

Response 4: We have added the clinical information in Table 1.

Comment 5: Was Fazekas score, or better, WMH volume used in any of the analyses. Since WMH is likely to be related to cognition, it would be pertinent to use this in analyses as it could independently explain some of the variance in MMSE score.

Response 5: We added the data pertaining to CMBs and the Fazakas score and analyzed. We have revised the manuscript and tables accordingly.

Comment 6: The sample size in two of the groups is very small, potentially biasing some of the results. This should be acknowledged in the discussion.

Response 6: Thank you for this pointing this out. Per your suggestion, we have mentioned it as a limitation in the Discussion.

Lines 308-311

This study has several limitations. First, this study was a retrospective single-center study. The sample size and selection bias were limitations. We divided the patients into four groups according to the ABI and baPWV; however, the sample size of these two groups was very small, potentially biasing some of the results.

Comment 7: Fazekas score for each group should be reported in the demographics table.

Response 7: Per your suggestion, we have added the Fazekas score in the Tables.

Comment 8: In the discussion, the authors mention that baPWV is preferable, but cfPWV has been established as a more robust measure and is the gold standard in the literature. It is also non-invasive. These caveats of baPWV should be acknowledged in the discussion.

Response 8: Thank you for the suggestion. We have added the following text:

Lines 280-282

The baPWV is measured between two sites along the arterial system and is preferred as it is easy to perform; however, cfPWV has been established as a more robust measure and is the gold standard.

Reviewer #2: The manuscript was substantially revised.

Comment 1: “exclud”→”exclude”

In line 72, “…Alzheimer’s pathology. Therefore, we tried to exclud cases with cerebral microbleeds…” should be changed as “…Alzheimer’s pathology. Therefore, we tried to exclude cases with cerebral microbleeds…”?

Response 1: We apologize for the typographical error. We have fixed the error.

Comment 2: p=0.10

In the methods, the authors described that “a value of p = 0.10 was used to indicate statistical significance.” Is there any literature that could refer to defend the choice of p=0.10?

Response 2: For performing the multivariate analyses, we selected the significant factors from the univariate analyses using the cut-off value of p=0.10. In order to select the optimal number of co-variates, we used the cut-off value of p=0.10. There is no fixed rule, but p=0.10 is sometimes used.

Lines 141-143

Univariate analysis was used to investigate the association of MMSE scores with several factors, and a value of p = 0.10 was used to indicate statistical significance for multivariate analysis.

---

## [Decision Letter · Decision Letter 2]

17 Jan 2022

PONE-D-21-22076R2Association of ankle-brachial index with cognitive decline in patients with lacunar infarctionPLOS ONE

Dear Dr. Nakamori,

Thank you for submitting your manuscript to PLOS ONE. After careful consideration, we feel that it has merit but does not fully meet PLOS ONE’s publication criteria as it currently stands. Therefore, we invite you to submit a revised version of the manuscript that addresses the points raised during the review process.

Minor revisions are necessary before acceptance.See the Reviewer's comments and respond them appropriately.

We look forward to receiving your revised manuscript.

Kind regards,

Masaki Mogi

Academic Editor

PLOS ONE

Journal Requirements:

Reviewers' comments:

Reviewer's Responses to Questions

**Comments to the Author**

1. If the authors have adequately addressed your comments raised in a previous round of review and you feel that this manuscript is now acceptable for publication, you may indicate that here to bypass the “Comments to the Author” section, enter your conflict of interest statement in the “Confidential to Editor” section, and submit your "Accept" recommendation.

Reviewer #1: All comments have been addressed

Reviewer #2: All comments have been addressed

2. Is the manuscript technically sound, and do the data support the conclusions?

Reviewer #1: Yes

Reviewer #2: Yes

3. Has the statistical analysis been performed appropriately and rigorously? 

Reviewer #1: Yes

Reviewer #2: Yes

4. Have the authors made all data underlying the findings in their manuscript fully available?

Reviewer #1: Yes

Reviewer #2: Yes

5. Is the manuscript presented in an intelligible fashion and written in standard English?

Reviewer #1: Yes

Reviewer #2: Yes

6. Review Comments to the Author

Reviewer #1: (No Response)

Reviewer #2: The manuscript was substantially revised, however several questions have been raised.

Comment 1: p=0.10

In the methods, the authors described that “Univariate analysis was used to investigate the association of MMSE

scores with several factors, and a value of p = 0.10 was used to indicate statistical significance for multivariate analysis. Subsequently, multivariate analysis was performed with selected factors determined from univariate analysis. " This sentence should be changed something like to "Univariate analysis was used to investigate the association of MMSE scores with several factors. Subsequently, multivariate analysis was performed to estimate and test the independent effects of selected factors on MMSE score. Each of those factors was determined from univariate analysis if the p value was 0.1 or less."

Comment 2: Power analysis

The authors have already declared that "Based on an alpha level of 0.05 and power of 0.80, we estimated that we would require at least 128 participants." However, in the discussion, the sentence "the sample size and selection bias were limitations." was presented. I could not understand this discrepancy.

Comment 3: Bonferroni correction and Tukey’s honestly significant difference (HSD) test

It is hard to understand the sentence "For multiple comparisons, the data were analyzed using one-way analysis of variance (ANOVA), and Bonferroni correction and Tukey’s honestly significant difference (HSD) test were performed." This should be changed to "For multiple comparisons, the data were analyzed using one-way analysis of variance (ANOVA), followed by post-hoc Tukey HSD test, with Bonferroni correction."

7. PLOS authors have the option to publish the peer review history of their article (what does this mean?). If published, this will include your full peer review and any attached files.

Reviewer #1: No

Reviewer #2: **Yes: **Michiaki Nagai

---

## [Author Response · Author response to Decision Letter 2]

17 Jan 2022

Thank you for reviewing our manuscript. We appreciate your comments and suggestions. We have revised the manuscript in accordance with your comments. Our point-by-point responses to the comments are presented below.

Response to the reviewers

Reviewer #2: The manuscript was substantially revised, however several questions have been raised.

Response: We appreciate your comments and advice. We have responded to each of your comments below and denoted the corresponding revisions in the manuscript in red color.

Comment 1: p=0.10

In the methods, the authors described that “Univariate analysis was used to investigate the association of MMSE　scores with several factors, and a value of p = 0.10 was used to indicate statistical significance for multivariate analysis. Subsequently, multivariate analysis was performed with selected factors determined from univariate analysis. " This sentence should be changed something like to "Univariate analysis was used to investigate the association of MMSE scores with several factors. Subsequently, multivariate analysis was performed to estimate and test the independent effects of selected factors on MMSE score. Each of those factors was determined from univariate analysis if the p value was 0.1 or less."

Response 1: We appreciate your helpful comment. According to your suggestion, we have revised the manuscript as follows.

Lines 141-144

Univariate analysis was used to investigate the association of MMSE scores with several factors. Subsequently, multivariate analysis was performed to estimate and test the independent effects of selected factors on MMSE score. Each of those factors was determined from univariate analysis if the p value was 0.10 or less.

Comment 2: Power analysis

The authors have already declared that "Based on an alpha level of 0.05 and power of 0.80, we estimated that we would require at least 128 participants." However, in the discussion, the sentence "the sample size and selection bias were limitations." was presented. I could not understand this discrepancy.

Response 2: We apologize for the confusing expression. We revised as follows.

Line 309

The selection bias was limitation.

Comment 3: Bonferroni correction and Tukey’s honestly significant difference (HSD) test

It is hard to understand the sentence "For multiple comparisons, the data were analyzed using one-way analysis of variance (ANOVA), and Bonferroni correction and Tukey’s honestly significant difference (HSD) test were performed." This should be changed to "For multiple comparisons, the data were analyzed using one-way analysis of variance (ANOVA), followed by post-hoc Tukey HSD test, with Bonferroni correction."

Response 3: We appreciate your helpful comment. According to your suggestion, we have revised the manuscript as follows.

Lines 146-148

For multiple comparisons, the data were analyzed using one-way analysis of variance (ANOVA), followed by post-hoc Tukey’s honestly significant difference (HSD) test, with Bonferroni correction.

---

## [Decision Letter · Decision Letter 3]

21 Jan 2022

Association of ankle-brachial index with cognitive decline in patients with lacunar infarction

PONE-D-21-22076R3

Dear Dr. Nakamori,

We’re pleased to inform you that your manuscript has been judged scientifically suitable for publication and will be formally accepted for publication once it meets all outstanding technical requirements.

Kind regards,

Masaki Mogi

Academic Editor

PLOS ONE

No further comment.

Additional Editor Comments (optional):

Reviewers' comments:

Reviewer's Responses to Questions

**Comments to the Author**

1. If the authors have adequately addressed your comments raised in a previous round of review and you feel that this manuscript is now acceptable for publication, you may indicate that here to bypass the “Comments to the Author” section, enter your conflict of interest statement in the “Confidential to Editor” section, and submit your "Accept" recommendation.

Reviewer #2: All comments have been addressed

2. Is the manuscript technically sound, and do the data support the conclusions?

Reviewer #2: Yes

3. Has the statistical analysis been performed appropriately and rigorously? 

Reviewer #2: Yes

4. Have the authors made all data underlying the findings in their manuscript fully available?

Reviewer #2: Yes

5. Is the manuscript presented in an intelligible fashion and written in standard English?

Reviewer #2: Yes

6. Review Comments to the Author

Reviewer #2: The manuscript was well revised. I have checked that the authors responded very well to our questions.

7. PLOS authors have the option to publish the peer review history of their article (what does this mean?). If published, this will include your full peer review and any attached files.

Reviewer #2: **Yes: **Michiaki Nagai

---

## [Editor Report · Acceptance letter]

25 Jan 2022

PONE-D-21-22076R3 

Association of ankle-brachial index with cognitive decline in patients with lacunar infarction 

Dear Dr. Nakamori:

I'm pleased to inform you that your manuscript has been deemed suitable for publication in PLOS ONE. Congratulations! Your manuscript is now with our production department. 

Kind regards, 

on behalf of

Dr. Masaki Mogi 

Academic Editor

PLOS ONE